# Videographic, Musical, and Linguistic Partnerships for Decolonization: Engaging with Place-Based Articulations of Indigenous Identity and *Wâhkôhtowin*

Joanie Crandall 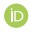

Faculty of Education, Yorkville University, Fredericton, NB E3B 3H4, Canada; jcrandall@yorkvilleu.ca

**Abstract:** N'we Jinan, a group of young Indigenous artists who run a mobile production studio and an integrative arts studio, travel to different Indigenous communities, where they support youth in writing and recording music that involves the local community. N'we Jinan employs social media to articulate and protect Indigeneity through the sharing of Indigenous music videos, empowering youth to resist continued colonization. These videos serve to create a sense of connection in Indigenous communities in Turtle Island (Canada) as well as offer a means by which non-Indigenous listeners can learn about contemporary Indigenous cultures. Viewed in conjunction with Nunavut's *Inuit Qaujimajatuqangit* and the Northwest Territories' *Dene Kede* and *Inuuqatigiit*, which provide a framework of traditional knowledge, values, and skills specific to Indigenous communities in the Canadian Arctic, the texts implicitly invite non-Indigenous listeners' engagement in social justice activism as settler allies. The texts invite listening to and viewing the empowering songwriting and recording practices through the lens of social justice and *wâhkôhtowin* or kinship relations, which involves walking together (Indigenous and settler) in a good way and engaging with Bourdieu's influential framework of cultural capital. The themes explored in the songs include cultural identity, language, and self-acceptance. The empowering songs of N'we Jinan are place-based articulations of identity that resist coloniality and serve as calls to action, creating embodied videographic, musical, and linguistic partnerships that serve as important articulations of Indigenous identity and which promote the decolonization of reading and listening practices and, by extension, education.

**Keywords:** Indigenous videography; Indigenous music; Indigenous language; cultural identity; social media; social justice; Inuit Qaujimajatuqangit; Dene Kede; Inuuqatigiit

## 1. Partnerships for Decolonization

*Okiskinohamawakanew Daylon, Devon, ekwa Joseph. Kîhtwâm ka-wâpamitin.*[1]

N'we Jinan is an educational organization that runs a Mobile Production Studio (MPS) and whose artist members teach skills in sound recording, music production, songwriting, and performance to youth of Indigenous[2] ancestry (N'we Jinan 2014, 2022a). Known simply as "N'we Jinan",[3] which translates from Eastern Cree (James Bay) as "We Live Here", the organization has, at the time of writing this article, supported over 900 Indigenous youth across 70 communities in becoming N'we Jinan artists. New media[4] texts produced from collaborative workshops organized by N'we Jinan MPS are available through numerous modalities: the N'we Jinan website (N'we Jinan Family of Sites 2022); Facebook (N'we Jinan 2022b); Instagram (N'we Jinan 2022c); Twitter (N'we Jinan 2022f); Soundcloud (N'we Jinan 2022e); iTunes (N'we Jinan 2022d); and YouTube (N'we Jinan 2022g). The youth artists engage in the supported interweaving of Indigenous melodies, drumming, language, and vocables with English-language singing and elements of spoken word hip-hop. The resulting videography is grounded in and visually linked to the land and traditional practices, often showing the youth artists singing and drumming in places of cultural significance. The videography renders Indigenous youth experiences accessible beyond

the space of the local community, creating connections through cyberspace to Indigenous and non-Indigenous audiences who have access to devices and, via the internet, YouTube.

Through intertwining cultural forms, the videography of N'we Jinan actively resists the colonizing gaze. N'we Jinan's lyrical and visual texts provide opportunities to engage in decolonization through alternative interpretive approaches and to subvert Eurocentric interpretive approaches by enacting an agentic social justice lens. Links can be made between the videos and accessible Indigenous Knowledge articulated in Anishinaabe Seven Teachings (Tacan 2022), *Inuit Qaujimajatuqangit: Education Framework for Nunavut Curriculum* (Nunavut Department of Education 2007),[5] *Dene Kede Education: A Dene Perspective* (Northwest Territories Department of Education, Culture and Employment 1993, 2002, 2003, 2004),[6] and *Inuuqatigiit: The Curriculum from the Inuit Perspective* (Northwest Territories Department of Education, Culture and Employment 1996)[7] through *wâhkôhtowin*[8] (Donald 2021) in ways that promote and enact decolonizing reading, viewing, and interpretative practices. N'we Jinan's body of work, I argue, offers rich opportunities to decolonize engagement with new media.

## 2. Positionality and Methodology

Before analyzing how the videos can be used as educational tools of decolonization, I would like to pause to clarify my methodology and positionality, as I will be interpreting N'we Jinan's videography and lyric texts through an interpretive lens that is framed by my experiences of living and working in Indigenous communities. I am non-Indigenous, and as I was writing this article, I was living and working on the unceded territories of the Wəlastəqwiyik and the Mi'kma'ki. I acknowledge my position as being an explicitly "unreliable narrator" (Chadderton 2012, p. 375) and the gaps and limitations of my subjectivities (L. Anderson 2006), although I am invested in maintaining meaningful and mutually supportive connections with Indigenous people alongside whom I have lived and worked.[9] I would also like to draw attention to the Indigenous epistemologies that inform my analysis of N'we Jinan's videos. *Wâhkôhtowin* (Donald 2021) is the "enmeshment within kinship relations that connect all forms of life" (p. 55) and is concerned with "how to conduct yourself as a good relative" (p. 58). Listeners/viewers can draw upon *wâhkôhtowin* (Donald 2021) in both literary interpretation and educational pedagogy. Listeners/viewers can also seek to connect to Indigenous knowledge through the educational framework of the *Inuit Qaujimajatuqangit* (2007) or the curriculum of the *Dene Kede* (Northwest Territories Department of Education, Culture and Employment 1993, 2002, 2003, 2004) and *Inuuqatigiit* (1996), which are exemplars of traditional knowledge, values, and skills. Daniel Heath Justice (2018) argues that analysis of Indigenous texts (which I extend to my analysis of new media in this article) requires "finding common ground that honours justice, embraces the truths of our shared history, and works for better futures takes courage and imagination—but most of all . . . love" (p. 179). As a result, analyses that value kindness and caring for all our relations (not only those immediately biologically connected to us) can begin repairing relationships between Indigenous and non-Indigenous people. This essay aims to perform an analysis of the music and music videos shared by N'we Jinan artists through seeking common ground as a way forward in reconciliation.

I also wish to address the complexity of postcolonialism (Hutcheon 1989; Said 1994; Slemon 1990) as related to Indigenous people in what is now known as Canada. According to First Nations, Métis, and Inuit perspectives, the idea of "post"-colonial is problematic (I. Anderson 2003; King 2004; Maracle 2004) since access to clean water, adequate housing, health care, and education continue to evoke colonial oppressions and hierarchies of racialized nation states, not their supersession. As a marker of my acknowledgement of these ongoing challenges (among others) and to demonstrate my respect for Indigenous cultures and the resiliency of Indigenous people, I employ "anti"-colonial and "de"-colonial instead of "post"-colonial.[10] Further, I employ methodologies grounded in social justice pedagogy since social justice is also integral to Critical Hip-Hop Pedagogy (Akom 2009; Kelly 2020; Ringsager and Madsen 2022). The videos of N'we Jinan can be examined as

part of a social justice pedagogical strategy that De Korne (2021) refers to as "language activism" (p. 1), which recognizes sites of contention and intersectionality. Similarly, Clarke and Bird (2021) argue that "Art and music are universal languages, but they may have specifically local dialects. The sense and meaning of visual representation can vary widely among cultures, groups, and historical periods" (p. 105).[11] Hence, it is necessary to engage with Indigenous languages and dialects, art forms, and music as specific iterations of their cultural context.

The work of N'we Jinan artists suggests that musical narratives of self-empowerment and resistance are compelling topics for Indigenous youth. The videography enacts counter-hegemonic imperatives of anti-colonialist agency (Cordes and Sabzalian 2020; Rice and Mündel 2018). In teaching hip-hop music, including that of N'we Jinan artists, the knowledge and perspectives of Indigenous youth must inhabit a central place in pedagogy (Kruse 2020, p. 155). N'we Jinan MPS' videography demonstrates the power of Indigenous youth in enacting what Janet Loebach et al. (2019) refer to as "digital storytelling" (p. 283).[12] Through the musical narratives of N'we Jinan's videos, Indigenous youth share moments of insight into their hopes, dreams, fears, struggles, challenges, successes, and visions for a better world through better relationships, which they evoke through powerful imagery, melodies, and lyrics. The youth artists create digital stories of their experiences as well as their aspirations.

N'we Jinan artists' videos are also helpful for educators who want to engage in dialogue promoting decolonization and other topics of social justice in their classrooms and who wish to employ Indigenous resources in so doing. Indigenous resources, such as the *Dene Kede* (Northwest Territories Department of Education, Culture and Employment 1993, 2002, 2003, 2004), *Inuuqatigiit* (1996), and *Inuit Qaujimajatuqangit* (2007), which are organized around relationships and seek to support learners, can be used to deepen dialogue around N'we Jinan's videos in the classroom. The *Dene Kede* aims to instruct learners "to become capable [which] means having responsible, skillful and respectful relationships with the spiritual world, with the land, with other people and with themselves" (Northwest Territories Department of Education, Culture and Employment 1993, p. 4). Familiarity with the *Dene Kede* helps audiences to understand the impetus of N'we Jinan's videography in terms of the responsibilities and respectfulness demonstrated in a plurality of relationships with traditions, place, community members, and with the self as a spiritual being. Similarly, the very title of the *Inuuqatigiit* (1996) evokes the deep meanings of "Inuit to Inuit, people to people, living together . . . This is the foundation of the curriculum: a unity of Inuit philosophy for the benefit of the children, teachers, schools and communities" (A.3). Similar to N'we Jinan MPS' body of work, the *Inuuqatigiit* (1996) serves as an embodiment of centralizing and respecting a multiplicity of strengths (G.30) and its core meaning is represented as constructed, such as the inuksuk, on a strong foundation of collaborative work to create a better future (C.16). These approaches and curricula are useful for social justice educators wishing to employ decolonizing approaches to interpreting N'we Jinan MPS' videography and youth artists' song lyrics.

## 3. Representations of Indigeneity in N'we Jinan Artists' Videography

Both the linguistic and visual markers of N'we Jinan videos cue YouTube audiences to the diversity of Indigenous languages, cultures, and geographical contexts, presenting empowering articulations of identity that resist coloniality. N'we Jinan videos introduce viewers to Indigenous youth artists in the context of their home community, the location of which is identified explicitly in the opening moments of each song. Figures 1 and 2 below illustrate the beautiful, complex, and deeply respected Indigenous traditions of the home community that is honoured through sharing by Indigenous youth artists in the videos. In integrating a multiplicity of representations of the Indigenous community and cultural forms, the youth in Aklavik assert in "Never Say Die"[13] that "the culture will bring us home" (N'we Jinan Artists 2023b). Echoing the *Inuuqatigiit* (1996), intended to "Encourage pride in Inuit identity to enhance personal identity" (A.5), the Aklavik youth demonstrate

through their lyrics their deep-rooted connection to their culture and their ability as they grow to "make good decisions" (C.15) for themselves. Throughout the song, the youth artists of the Gwìch'in and Inuvialuit communities appear as a collective, in small groups, and individually, recording and performing the lyrics that they contributed, celebrating individual voices while simultaneously resisting the privileging of individual voice over community (N'we Jinan Artists 2023b).

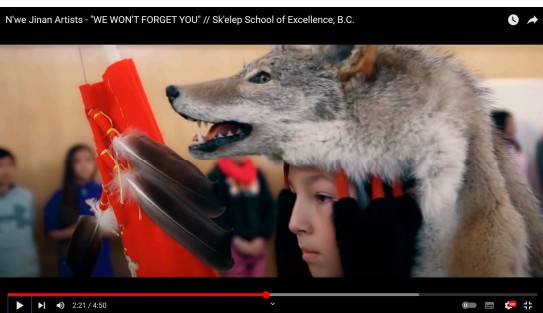

**Figure 1.** N'we Jinan, "We Won't Forget You", YouTube, 19 December 2017, 2:21, video, https://youtu.be/u0YYkvIWbng (accessed on 5 March 2023).

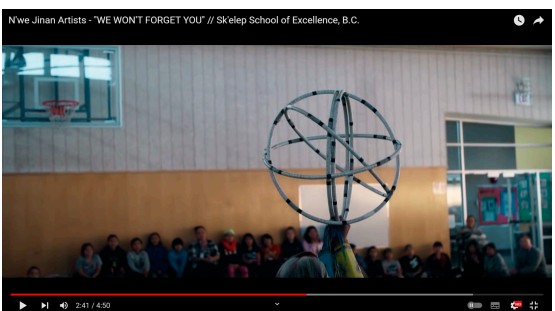

**Figure 2.** N'we Jinan, "We Won't Forget You", YouTube, 19 December 2017, 2:41, video, https://youtu.be/u0YYkvIWbng (accessed on 5 March 2023).

The videography of N'we Jinan MPS is, each time, a form of culturally specific production that articulates the experiences of Indigenous youth artists grounded in their cultures and places. The approach focuses on conscious attention to context, "situating the knowledge we produce in ethnographic research as located, partial, and subjective" (Chadderton 2012, p. 376); further, "The cultural production approach puts the emphasis on the particular contexts, people, materials, and relationships in which these kinds of practices occur" (Recharte 2019, p. 75). Some cultural practices are rendered visible through the videography, with the youth artists sharing videos of themselves engaging in smudging and traditional dancing, for example, or moving through their home and school and community spaces, gathered around fires or significant landscape formations, or drawing attention to culturally specific signifiers, such as totems, tipis, hoops, and traditional regalia. The compelling videography simultaneously underscores the diversity of cultures and the commonality of the issues that impact Indigenous youth across Canada.

N'we Jinan MPS empowers Indigenous youth to resist continued colonization by celebrating culture, language, place, and experience through music and videography. The liberatory expression of new media as a decolonizing and Indigenizing reclamation of place, culture, and experience evokes numerous tensions and complexities. Because N'we Jinan MPS mentors travel, the music and videography offer insight into Indigenous communities across Canada and render visible Indigenous communities' deep connection to the landscapes which have shaped them.[14] In Figures 3 and 4 below from "Break the Silence" (N'we Jinan Artists 2017a), for example, the youth artists self-locate on the road within their community while also celebrating their connection to the land by recording

their presence overlooking the water. The centrality of the landscape is important in using these videos in classroom contexts for, as Linda Wason-Ellam (1991, 2005, 2010) underscores, engaging in place-based learning is an invaluable approach to authentic learning. Mercedes Peters (2019), too, affirms the centrality of place to Indigenous people. Contextualizing the visual in the videography through identity and place also supports culture-based arts integration, which contributes to the "debunking monocultural assumptions about 21st-century Indigenous life by privileging place-specific conceptions" of Indigenous identity (Bequette 2014, p. 220). The music articulates and protects the multifaceted expressions of culture, providing evidence of deeply held connections.

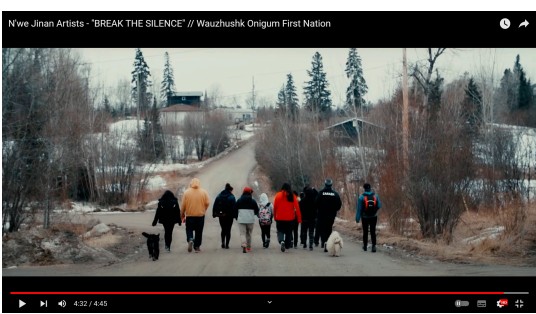

**Figure 3.** N'we Jinan, "Break the Silence", YouTube, 24 April 2017, 4:32, video, https://youtu.be/30PS-h6yKvk (accessed on 5 March 2023).

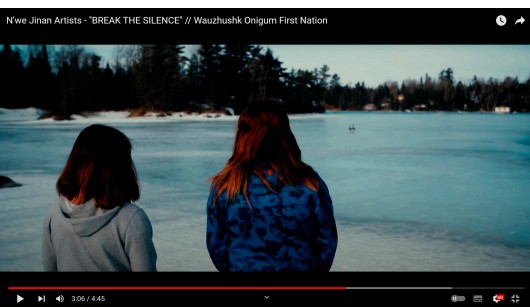

**Figure 4.** N'we Jinan, "Break the Silence", YouTube, 24 April 2017, 3:06, video, https://youtu.be/30PS-h6yKvk (accessed on 5 March 2023).

The complex interrelationship of culture, language, and place is alluded to visually in ways that parallel and complement the linguistic forms. The visuals enact both documentation and representation of Indigenous culture, music, and language, and the audience is invited to follow the path wended by the youth artists, which renders visible the diverse landscapes inhabited by Indigenous people. The videography of the youth artists recording their singing is partnered with images, both localized and inclusionary, through drone recording that reinforces the unequivocable presence of the film as a tool of resistance. Indigenous ways of knowing are conveyed through sophisticated drone techniques; the videography illustrates the high production value of multiple angles and multiple shots incorporated into the visual narrative. That the participants of the videos have been effectively coached in resisting digital re-inscriptions of colonial production is evidenced through advanced video production. The youth artists alternate singing into the landscape, facing into the camera, and into the recording devices made visible in other shots where the local school's incorporation of Indigenous images is rendered an authentic mise-en-scène. N'we Jinan MPS' videography, which follows the artists walking through the landscape or paddling through the waters, serves to evoke multiple interpretive paths grounded in Indigenous ways of knowing. N'we Jinan MPS' videography emphasizes the embodied agency of Indigenous youth in negotiating the spaces they traverse individually and collectively, engaging in the process of mapping their journeys through the metaphor

of space. The youth artists in the video repeatedly cross, query, and reinforce borders, claiming, reclaiming, and reconfiguring the physical and liminal spaces that they inhabit.[15] Through the praxis of connectivity interwoven into the visual and linguistic elements of the videography, the youth artists engage in the process of linking localized storied production to the impetus of the new media. In the videos, the youth artists are empowered to resist dominant public narratives through the praxis of connectivity within self-chosen locations. As such, the videos lend themselves to decolonizing approaches to interpretation.

The sounds that accompany the compelling visuals communicate the self-empowerment of Indigenous youth, as many songs incorporate traditional drumming as well as Indigenous language and vocables. As Figures 5 and 6 below from the music videos "Home to Me" (N'we Jinan Artists 2016a) and "I Believe" (N'we Jinan Artists 2014a) capture in visual terms, Indigenous drumming—individually or as a group—is a central, unifying element in the music produced and enacts self-determination and, as such, "a compelling channel for spatial decolonization" (England 2019, p. 10). N'we Jinan artists underscore the possibilities inherent within the subversive social justice potential intrinsic to new media. As an example, Secwepemc youth artists in "We Won't Forget You" embrace the opportunity of new media while invoking and reclaiming language and cultural practices, asserting that "language is taught through drumming and singing" (N'we Jinan Artists 2017b, Verse 9), drawing upon their mother tongue to assert, "Da-eh-mess-lep-eh-dem / We won't forget you / . . . / Tqelt Kukpi" (Chorus). The youth artists share moments of pride juxtaposed with deep pain, rendering evident how community efforts in language and cultural revitalization empower the youth.

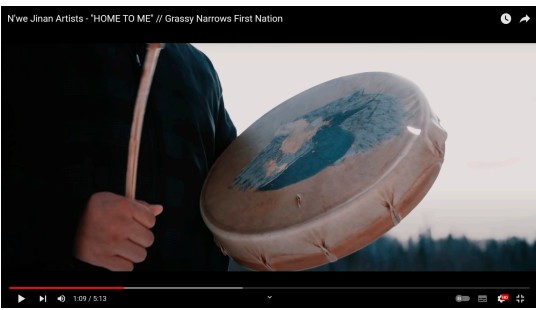

**Figure 5.** N'we Jinan, "Home to Me", YouTube, 18 March 2016, 1:09, video, https://youtu.be/EgaYz8 YWsO8 (accessed on 5 March 2023).

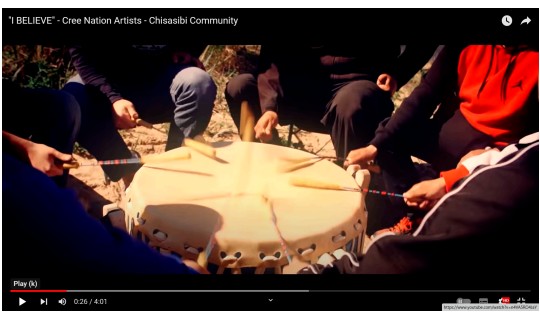

**Figure 6.** N'we Jinan, "I Believe", YouTube, 4 November 2014, 0:26, video, https://youtu.be/3RFT1 Pu0e28 (accessed on 5 March 2023).

Audiences of N'we Jinan artists are, thus, invited to engage with self-empowering Indigenous-created new media as learners of decolonizing practices rather than consumers. N'we Jinan artists help the audience begin taking steps toward "Cultural fluency . . . [which] involves approaches centred in self and other-awareness that shift old perceptions and facilitate multimodal communication" (LeBaron and Sarra 2018, p. 45). Laurel Smith (2010) points out that "greater access to communication technologies and the skills to

use them facilitate . . . cultural reworking of the world" (p. 271). Following Indigenous teachings, audiences must extract meaning for themselves. Further, as Juliet Hess (2017) notes, such work extends possibilities in moving music education in schools in the direction of social justice.

## 4. Linguistic Negotiation of Indigeneity in N'we Jinan Artists' Lyrics

Reclaiming Indigenous language as a means of self-empowerment underpins many of N'we Jinan artists' self-representation choices. Significantly, N'we Jinan's introductory self-titled video is completely in Cree (N'we Jinan Artists 2014b),[16] suggesting that subsequent works that include Indigenous language may be part of a process actively encouraged by N'we Jinan MPS and, further, is supported by local language keepers during the MPS' time in communities. In promoting the Indigenous language through lyrics and vocables, N'we Jinan MPS and youth artists together reinforce the intrinsic value of reclaiming and celebrating the Indigenous language. As the Sámi musician Sofia Jannok powerfully asserts, "art and culture and music give a more fair and true image of reality because it is told through the eyes of the ones who experience it" (Blomqvist 2016, para. 11). Subsequent N'we Jinan artists' songs and the provided lyrics share stories that engage in code-switching, in which the artists move fluidly between one language and another (Gordon and Williams 1998). As J. Edward Chamberlin (2003) states, "The idea that we live our lives in language and that we understand the world differently because we speak different languages goes back a long way" (p. 17). One can listen to a multiplicity of Indigenous languages through N'we Jinan's videos on YouTube, each grounded in its own specific geographical place, each celebrating the resilience of Indigenous communities. The reclamation of language and song renders the videography particularly impactful and lends itself to reading and viewing through a decolonizing lens.

N'we Jinan youth artists present counter-stories of strength and resilience through both visual and linguistic cues that offer markers for decolonizing reading and viewing practices. As Teresa Strong-Wilson (2009) argues, "decolonizing the imagination [can also proceed] through story" (p. 116). The poetic impulses of the lyrics that interweave with the powerful videography and soundscapes invite the performance of close reading through a social justice lens that engages with Indigenous resources in enacting interpretation. Close reading can be performed through exploring not only the imagery but the links to and performances of Indigenous ways of knowing. Notably, the most easily accessible to non-Indigenous audiences is the Seven Sacred Teachings: Love, Humility, Honesty, Wisdom, Courage, Truth, and Respect (Tacan 2022). "Creespect" (N'we Jinan Artists 2015), for example, is a portmanteau of Cree and the Teaching of Respect (Tacan 2022): "Respect the lands / . . . / Proud of my people" (Verse 3), and in empowering terms, "you are growing proud and humble / . . . / One step at a time, we're striving for the better" (Verse 6). The parallel can be seen in the *Inuit Qaujimajatuqangit* (2007) value *Ikpiguhungniq*—Respect.[17] Although at the time this paper was being written, N'we Jinan MPS had not yet reached Nunavut, similarities in values among the Inuit and Inuvialuit with more southern Indigenous cultures would help educators find an entry point to engage learners in reflective dialogue about the many Indigenous cultures represented in the N'we Jinan videos. For example, there is an opportunity to discuss the value of Strength (Tacan 2022) in the assertion by the Gwìch'in youth who perform "Everlasting" (N'we Jinan Artists 2023a), "I am proud of where I come from/Singing loud, we're still strong"[18] (Chorus). Aligning with the *Dene Kede* for Grade 9, the youth artists enact "opportunities to explore and experience many aspects of the culture so they can better know their own interests or strengths" (Northwest Territories Department of Education, Culture and Employment 2004, p. 20). The song "Everlasting" is a demonstration of collaboration; the YouTube page explains that the song "was created by a group of youth from Fort McPherson and Tsiigehchic [sic]" (N'we Jinan Artists 2023a), communities in relatively close proximity in the Beaufort Delta. Together, Gwìch'in youth artists from the two communities embody *Dene Kede*'s guidance to search for their talents and strengths.

The lyrics shared in the N'we Jinan videos that are posted to YouTube are public in their accessibility and render collaborative work available for diverse audiences, particularly encouraging engagement through a decolonizing listening/viewing practice. In "We are Strong" (N'we Jinan Artists 2022b), Métis youth artists assert, "We are different people / But we are all equal" (Verse 1). Similarly, the Anishinaabemowin youth artists in "Break the Silence" (N'we Jinan Artists 2017a) demonstrate the Sacred Teaching of Courage (Tacan 2022) in speaking the truth about their experience of being subjected to colonial practices yet their willingness to share the traditional teachings of their culture to help create a more equitable and fair society, "Our culture's disrespected / . . . / The teachings, can you share them?" (Verse 3). In the chorus that follows, the youth artists invite listeners to "Come and find peace with us / And break the silence with love." As "Break the Silence" (N'we Jinan Artists 2017a) closes, they proclaim,

> We, children, are the medicine, the bringers of tradition
> Keepers of the peaceful, sacred life that we've been given
> We're the bridge of two paths, the voice of our spirit
> The forgivers of a nation, we are one voice singing. (Verse 13)

In parallel terms, in "Don't Give Up" (N'we Jinan Artists 2023c), the Inuvialuit youth artists affirm the strength of their desire to "keep the culture growing" (Verse 2), enacting the *Inuuqatigiit* goal to connect past and present (1996, A.5). The lyrics, which are compelling in their artful simplicity, enact agentic resistance.

Engaging with N'we Jinan videos as new media reclaiming language and enacting cultural revitalization renders the complexities of decolonizing textual engagement for non-Indigenous audiences more immediately apparent. Engaging implicates audiences in considering the intersection of reading, viewing, and interpretive contexts. N'we Jinan artists' work enacts a place-based embodiment of new media that invites social justice approaches. As such, it requires the conscious acknowledgement and consideration of how one's language shapes notions of truth, self-representation, and interaction with the world. Similar to Halluci-Nation, formerly A Tribe Called Red, the work of N'we Jinan navigates "temporal boundaries by combining traditional and contemporary musical techniques . . . [with] messages of Indigenous nationhood and reassertions of Indigenous cultures . . . an act of creative contention against settler colonial cultural production" (Barker 2015, p. 59). Within the partially contextualized narratives and visual discourses and representations of cultures, places, and identities, disruptions and subversions of digital colonialisms occur, such as when the Cree youth artists assert in "When the Dust Settles" (N'we Jinan Artists 2016c), "You'll realize that we have something special" (Verse 5). Such narratorial assertions reinforce the centrality of culture to Indigenous experience. As in the work of Juneau-based rappers Arias Hoyle and Chris Talley, "location and language are showcased as repositories of cultural identification" (Balestrini 2022, p. 55). N'we Jinan videography, asserting cultural value through language and reinforcing such value through the accompanying images of place, serves to acknowledge in visible terms the need for the disruption of dominant stories through a multimedia approach to the retelling of contemporary Indigenous narratives.

In videographic partnership in the work of N'we Jinan artists, there is clear evidence of rap and hip-hop influences, genres demanding notice and positive change in response to social inequities that demand decolonizing interpretive practices. The rap and hip-hop influences embody and amplify linguistic and musical resistance (Clarke and Hiscock 2009; Sarkar and Winer 2006) within N'we Jinan's body of work, as code-switching is employed in tandem to engage, defy, and transcend identity borders (Sarkar et al. 2005). Camea Davis' (2021) articulation of the term protest is useful to consider here in the context of interpreting N'we Jinan's videography, "protest is the overt challenging or speaking out against an oppressive idea or practice in an attempt to create positive change" (p. 121). The expression of vernacular knowledge through the colonizer's language, which both inhibits forms of expression and renders other meanings more accessible across linguistic divides, effectively serves to co-opt the language of colonization. Linguistic performances that engage in code-switching, weaving in and out of the colonizer's language with Indigenous

language, actively resist and contest singular and negative narratives imposed historically by non-Indigenous people and draw attention to the importance of caring for the land and waterways that sustain everyone, whether they are immediately aware or not. Anishinaabe youth artists assert in "The River Flows" (N'we Jinan Artists 2018), "I will stay strong/Til everyone's listening/Singing for hope but crying for help, things don't change" (Verse 3). Similarly, in "Protect our Land" (N'we Jinan Artists 2022a), Tsleil-Waututh youth invite the audience to participate collaboratively in positive change, "Protect our land from the pollution / Hold my hand, let's find a solution" (Verse 1). Parallel to the resistances offered by N'we Jinan youth artists, there are noteworthy calls from Indigenous people in Australia for colonial powers to address the inequities created through oppressive treatment and environment-damaging practices in their country. In Baker Boy (2018), a collaboration of Yolngu rapper Baker Boy and Noongar rapper Dallas Wood, the following protest is offered to incite non-Aboriginal listeners to participate in creating change, "The day that they listen will be the day that I see a difference" (Verse 2), closing the song with the compellingly provocatory "If you don't wanna close the gap then close your gap" (Verse 3). As in the work of the artists of N'we Jinan, the push to create positive change is rendered both visible and audible in new media through the hybridization of traditional forms.

Many of the songs of N'we Jinan youth artists feature traditional drumming, singing, and vocables, literally and metaphorically grounding the music within the Indigenous community while weaving in rap and hip-hop rhythms and speech patterns within the colonizing language of English. N'we Jinan artists' linguistic choice to use the colonizer's language, in effect, acts as a counterpoint, creating a polyphonic aural experience. Exemplars of these choices occur in "Creespect" (N'we Jinan Artists 2015), with the introduction in Cree and the remainder of the song in English, in "Home to Me" (N'we Jinan Artists 2016a) with Anishinaabemowin vocables opening the melody, and in "Protect Our Land" (N'we Jinan Artists 2022a), in which Tsleil-Waututh vocables are honoured through repetition and in closing the song. The resilience demonstrated through the multiplicity of vocal resistance by N'we Jinan artists acts as a counter-public to the colonial state and historical violence of colonialism.

## 5. Connectivity and Access

The work of N'we Jinan embodies opportunities for subverting colonial interpretive practices and decolonizing and Indigenizing reading, listening, and viewing through conscious engagement with language, music, and videography that collaboratively serve agentic purposes and honour Indigenous subjectivities and new voices. N'we Jinan MPS supports authentic place-based learning opportunities, and there is deep pedagogical potential embedded in engaging with their work. N'we Jinan's videography demonstrates how, as LeBaron and Sarra (2018) argue, "aesthetic experience is inextricably linked to social transformation" (p. 4). N'we Jinan artists evoke active participation in meaning-making, offering compelling sites of convergence of Indigenous subjectivities and the anti-colonial struggle.

The work of N'we Jinan celebrates the resilience of Indigenous communities even as the youth artists navigate the inherent anxieties and tensions which accompany engaging with technology as a tool for resisting colonialism. Navigating the tensions requires youth to be drawing upon the tandem effort of the community in ways that echo the united efforts of rowing, as demonstrated below in Figure 7 from "Protect Our Land" (N'we Jinan Artists 2022a). The videography of N'we Jinan MPS, complementing the overlaid lyrical narratives, problematizes notions of inclusion and enclosure and the complexities of resisting gazes of consumption. The problematizing process can be read akin to the Idle No More movement, for "If Idle No More has demonstrated anything, it is that Indigenous peoples will not cease pursuing decolonization, nationhood, and social change because that is the condition and effect of their existence" (Barker 2015, p. 60). Employing technology as a tool of resistance to racism enacts, similar to the powerful impetus rendered evident through the Black Lives Matter movement, a praxis of digital allyship through which to

move toward racial justice (Clark 2019). Authentic dialogue may even occur, as Nikita Carney (2016) argues, through digital spaces because a Fanonian flattening of authority (Bhabha 2015) can occur through digital participation. Furthermore, engaging in dialogue around digital spaces and new media productions as tools of resistance also demands an expansion of Eurocentric academic frameworks to ensure "equal voice to all literacy performances, releasing students' imagination, creativity, and criticality to learn deeply" (García and Kleifgen 2020, p. 567). Such an approach thereby invokes Freire and Macedo's (2005) teacher–learner/learner–teacher dynamic. Indigenous youth artists, empowered to articulate their experiences in meaningful ways through their individual expression of language, culture, and place, create a digital space where non-Indigenous audiences can witness the cumulative effects of the resilience and resistance of Indigenous communities even if artists and audiences are geographically distant. Taken as a whole, the work of N'we Jinan performs and queries the notion of digital connectivity through embodying expressions of resistance.

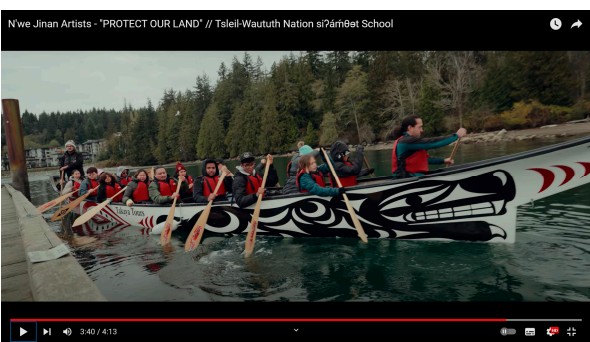

**Figure 7.** N'we Jinan, "Protect Our Land", YouTube, 7 December 2022, 3:40, video, https://youtu.be/kVWVSJKC0xs (accessed on 5 March 2023).

Through N'we Jinan MPS' productions, the audience of new media made available on YouTube is granted small glimpses into the lives of resilient Indigenous youth artists each time a new video is shared. Many of the youth artists who participate in N'we Jinan initiatives live in remote communities with limited opportunities available to them. However, rather than focusing on the limitations, N'we Jinan MPS serves to empower Indigenous participants through supporting, reinforcing, and celebrating their linguistic and cultural knowledge and the resilience of their communities. The youth artists engage in the reclamation of land through walking, reminding the audience that the land was stewarded by their ancestors long before settlers arrived at the shores of what is now known as Canada. Unfortunately, the N'we Jinan program, although it promotes resistance, is also delimited by economic access, which is not explicitly addressed on their sites. As with so many other educational opportunities, N'we Jinan MPS' ability to empower young Indigenous people, even before the pandemic, would have been affected by such constraints as weather-impacted travel concerns and limited accommodation spaces, which are ongoing challenges in remote and northern communities. Remote and northern communities exemplify the literal digital divide in Canada. The work also draws attention to the pressing need to bridge the digital divide that is an impediment to reconciliation. At the time this article was written, N'we Jinan had 17.5 K subscribers on YouTube and 163 videos; moreover, the most popular music video at the time of writing, "Home to Me" (N'we Jinan Artists 2016a), had been viewed over 655 K times since its posting in 2016 on N'we Jinan's YouTube channel. N'we Jinan continues to demonstrate its traction with audiences with songs and videos that have been viewed more than 15 million times. There is great potential for further studies to occur as Indigenous community partnerships grow, and the body of N'we Jinan work continues to expand. The work of N'we Jinan offers a means by which to bring more awareness to and celebrate the diversity of Indigenous cultures in Canada.

## 6. Conclusions

N'we Jinan MPS' decolonizing videographic, musical, and linguistic partnerships serve to create a sense of connection in communities of people with Indigenous ancestry in Canada as well as offer a means by which non-Indigenous audiences can respectfully explore themes of Indigenous cultural identity, language, self-acceptance, and resiliency through a decolonizing lens. Further, the work of N'we Jinan offers a multiplicity of opportunities to accept the implicit—and, at times, explicit—invitations to walk together through *wâhkôhtowin* (Donald 2021). Viewing N'we Jinan artists' work in conjunction with Indigenous Knowledge—such as the Seven Sacred Teachings (Tacan 2022), *Inuit Qaujimajatuqangit* (2007), *Dene Kede* (Northwest Territories Department of Education, Culture and Employment 1993, 2002, 2003, 2004), and *Inuuqatigiit* (1996), which provide exemplars of place-based traditional knowledge, values, and skills—empowers Indigenous youth to participate in articulations of identity that resist coloniality and invites non-Indigenous audiences to learn about Indigenous cultures. Moreover, N'we Jinan videography as new media offers important vantage points through which to decolonize viewing/listening/reading practices and, by extension, education. N'we Jinan's work is a call to action to right past and current systemic wrongs that create more than a digital divide. As suggested visually below in Figure 8 from "We are Medicine" (N'we Jinan Artists 2016b), there is, instead, a healing power in walking together through *wâhkôhtowin* (Donald 2021) with love.

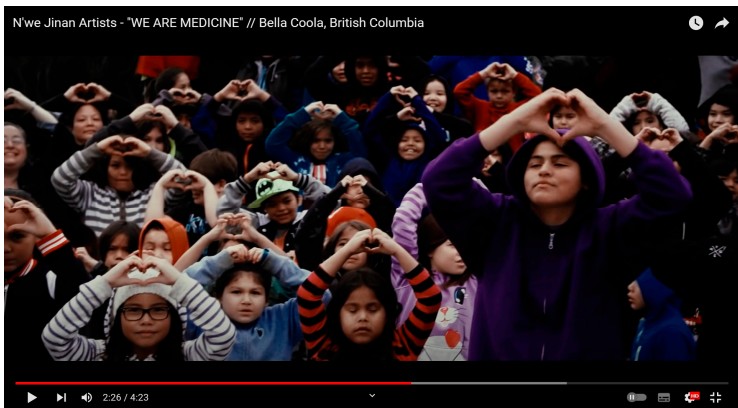

**Figure 8.** N'we Jinan, "We Are Medicine", YouTube, 7 December 2016, 2:26, video, https://youtu.be/VeWqgLLCef0 (accessed on 5 March 2023).

**Funding:** This research did not receive funding.

**Institutional Review Board Statement:** Not applicable.

**Informed Consent Statement:** Not applicable.

**Data Availability Statement:** No new data were created or analyzed in this study. Data sharing is not applicable to this article.

**Acknowledgments:** My thanks to organizers Rebecca Weaver-Hightower and Lorenzo Veracini for their helpful comments in the development of this article.

**Conflicts of Interest:** The author declares no conflict of interest.

## Notes

[1]  English translation: To Daylon, Devon, and Joseph, who also taught me. We will see you again. With the families' permission, I am dedicating this current work on decolonization and Indigenization to the young Nêhiyawak Cree men who are no longer with us.

[2]  Indigenous, in this article, refers to individuals and groups descended from people—which include First Nations, Métis, and Inuit people—who lived in the country known as Canada before settlers arrived.

3    The members of N'we Jinan are artists who travel to Indigenous communities. They support youth artist participants in collaborating in writing, performing, recording, and producing original songs (N'we Jinan 2022a). All merchandise profits feed the youth programs (N'we Jinan Artists 2014a). While the offices are located in Tiohtiá:ke (Montreal), the members of the mobile production team conduct their work in Indigenous communities and remotely, so it is useful to view the details provided in each of the YouTube videos to identify the individual N'we Jinan members who assist with the video, music production and mixing, and mastering as well as the community in which the video occurs and the individual N'we Jinan youth artists who participate in the video.

4    In exploring the videos in this article, I follow Lev Manovich's considerations of the disruption and redistribution of power embedded in past cultural categories (Manovich 1998) and Sonia Livingstone's definition of new media as already "familiar technologies" (Livingstone 1999, p. 61).

5    Future in-text references will refer to this document as the *Inuit Qaujimajatuqangit*. Educators are explicitly directed to incorporate this resource into their teaching in Nunavut classrooms. While N'we Jinan has not yet worked with Nunavummiut youth, I incorporate this document here to draw attention to it for its potential in terms of engaging with literature. I advocate for using locally-specific Indigenous resources and frameworks where possible and acknowledge that there are many unique differences between Indigenous cultures even when they share a language, and I also acknowledge from my experience as a teacher in Indigenous communities that sometimes a needed entry point to a text within a teaching context comes from identifying similarities and differences between the culture and language being studied and that of the learners. The *Dene Kede Education: A Dene Perspective Kindergarten—Grade 6* explicitly states an impetus to "Recognize similarities and differences between Dene and others" (Northwest Territories Department of Education, Culture and Employment 1993, p. 5). Similarly, the *Inuuqatigiit: The Curriculum from the Inuit Perspective* states that "the curriculum [has] something for people of many different backgrounds. We want to celebrate the similarities of all people, rather than differences" (Northwest Territories Department of Education, Culture and Employment 1996, p. 3). I draw upon these texts because of my first-hand experience using them in Arctic schools.

6    Future in-text references will refer to this set of documents as the *Dene Kede.* Educators are explicitly directed to incorporate these resources into their teaching in Northwest Territories classrooms.

7    Future in-text references will refer to this document as the *Inuuqatigiit*. Educators are explicitly directed to incorporate this resource into their teaching in Northwest Territories classrooms.

8    According to (Cree Dictionary Online 2023), *wâhkôhtowin* means "relationship" or "the state of being related to others". This, I believe, is a particularly important idea as part of reconciliation in educational contexts so that Indigenous and non-Indigenous learners, educators, instructional leaders, and administrators can walk together.

9    I have been privileged and honoured to support Nêhiyawak (Cree), Inuit, Inuvialuit, Gwìch'in, and Métis learners in a number of educational roles. I am grateful to each of the people who have supported and continue to support my learning journey through Indigenous languages and cultures both in person and online, formally and informally. This article represents my efforts at "being and becoming a good relative . . . [through] active and meaningful engagement—relatives aren't just static roles or states of being, but lived relationships" (Heath Justice 2018, p. 73). The work of N'we Jinan offers learning opportunities for educators like myself who are interested in social justice. Learning about how empowering linguistic and visual articulations of Indigenous identity resist coloniality is an important underpinning to teaching about social justice.

10    The narratives emerging from contemporary Indigenous, Inuvialuit, and Métis youth require interpretation that acknowledges the ongoing impact of colonialisms. Consciousness of colonialisms enacts an "an entry point into agency" (Tlostanova 2019, p. 165). The term "decolonizing" foregrounds Indigenous ways of knowing and doing (Mullen 2020) and can connect accessible traditional knowledge with potential pedagogical approaches (Thevenin 2022, p. 16). One of the ways in which I attempt to honour the gifts of learning about Indigenous cultures that I have received is through engaging with Indigenous resources and epistemologies (Battiste and Henderson 2000; Battiste et al. 2002; Donald 2021; Tuhiwai Smith 1999).

11    See also Craig Womack's (1999, 2005) arguments for cultural specificity in interpreting literary texts and Nassim Balestrini's (2022) call to research hip-hop artists and their music in ways that are contextualized within their specific location.

12    According to Janet Loebach et al. (2019), digital storytelling is a narrative methodology (p. 283) that is participatory (p. 284), supports participants in shaping their own narratives (p. 284), encourages creativity (p. 286), and can be captured through cell phone recordings as well as more expensive equipment (p. 294).

13    For more information on the town's motto, after which the song is named, see Cooper (2010).

14    A form of critical pedagogy for educators must be enacted that is cognizant of the impact of racialized and cultural context (Giroux 1992, 1999). It must be agentic, reflective, and reflexive and attend to how linguistic and visual elements enact selfhood and negotiate figured worlds (Holland et al. 1998; Urrieta 2007). Further, language, culture, and media representation require explicit engagement, particularly in the ways in which they disrupt, query, or reinforce experiential themes of survival (Wiggins and Monobe 2017).

15    N'we Jinan offers rich opportunities for forms of what Giroux (1992) refers to as border crossing. Negotiating border spaces and reflecting upon and engaging in the epistemic disobedience of pluriversality furthers decolonial thinking (Mignolo 2011, p. 63) and has the potential to initiate or further epistemic activism (Behari-Leak 2020, p. 19) as part of a social justice pedagogy.

16    The N'we Jinan website provides the opportunity for viewers/listeners to download the phonetically spelled Cree lyrics. The phonetic spelling is beneficial in rendering the sounds immediately accessible but can provide a challenge in seeking translation to access meaning via published Cree in language online/textual dictionaries, which tend to employ Standard Roman Orthography (SRO).

17    The poster *Ikpiguhungniq*, expressed in the Nattilingmiutut dialect of Inuktitut, represents Respect through the image of a caribou nibbling lichen under a bright Arctic sun. The image suggests the need to respect the cycle of life in the same manner as the caribou, to take only what one needs and to appreciate what is being granted from the land and the lives that give of themselves that people may also be sustained.

18    The *Inuit Qaujimajatuqangit* value *Ihumamut Akhuurniq*, expressed in the Nattilingmiutut dialect of Inuktitut, also explores the value of Strength in an Arctic context. The poster represents Strength through the image of a polar bear gazing off from a small ice floe, an image ever more powerful as the increasing impact of climate change becomes ever more evident in the north.

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
