# Peer review of "Videographic, Musical, and Linguistic Partnerships for Decolonization: Engaging with Place-Based Articulations of Indigenous Identity and Wâhkôhtowin"

_humanities, doi:10.3390/h12040072_

Round 1

Reviewer 1 Report

The article reports on the YouTube videos posted by a Canadian Indigenous organization known as N’we Jinan. Although the abstract promises to present the videos as vehicles for Indigenous youth empowerment and Indigenous decolonization though the development of cultural capital, the author provides little evidence of either.

Readers learn almost nothing about N’we Jinan other than they are “a group of young Indigenous artists who run a mobile production studio … [and] travel to different Indigenous communities” where they get Indigenous youth to perform in music videos. How many people comprise N’we Jinan? What genders? When was the organization formed? By whom, for what reasons? How is it organized? Is it a collective, an NGO, something else? We are led to believe that they are Cree – are they all Cree? Which Cree? What are their guiding principles? What are their funding arrangements? Are the youth paid for their performances? Do they receive training in video editing and production? Are the YouTube’s monetized? Who owns the copyrights to the videos? Where does N’we Jinan fit in the vast ecosystem of NGOs that purport to improve the lives of Indigenous youth?

How did the author collect data for the article? It appears that she watched the N’we Jinan YouTubes only. This is fine, but if that is all she did, she cannot make statements about what benefits the youth derive from participation. Did she interview members of N’we Jinan? Observe N’we Jinan’s work with any Indigenous youth? Speak to any youth who participated in a workshop? P. 5, lines 162-4 read: “The youths are the creators and subjects of the music and the accompanying video while members of the N’we Jinan mobile production studio and the elements of production remain behind the scenes.” If this is the case, in what ways does N’we Jinan subtly or explicitly direct the content and form of the videos? How does the author know that “the youths engage in authentic and meaningful modalities of expression and storytelling” (p.6, lines 211-12)?

The author jumps from theoretical construct (postcolonialism, decolonization, Indigenization, social justice pedagogy – where does Bourdieu come in?) to theoretical construct without fully engaging with any or connecting to her data. I would prefer that she limit herself to theory that she can demonstrate through the videos. What language is the term wâhkôhtowin? How widely is it employed by Indigenous people in Canada? The author makes extended reference to 2 Inuit terms (Inuit Qaujimajatuqangit & Inuuqatigiit) and one Dene term (Dene Kede). The last 2 appear to have been pulled from NWT curriculum materials, but it is not clear if or how they are institutionalized in actual Indigenous communities. The 3 terms are presented with what appear to be publication dates, but are not listed in the references. Also missing is any meaningful engagement with the vast literature on “new media.” For example, what exactly does the author include as “new media”? What are the affordances of digital media that serve the needs of Indigenous youth? How are professionally produced YouTube videos alike or different from diy TikToks? How is the fact that these are “social media” relevant? Or not?

Why does the author make a distinction between Indigenous and Metis youth? Is she claiming that Metis are not Indigenous? Or not Indigenous enough?

Far too much jargon: "In the videos, as in the documented values and knowledges of Inuit Qaujimajatuqangit, (2007), Dene Kede (1993), and Inuuqatigiit (1996) the youth are empowered in resisting dominant public narratives through the praxis of connectivity within self-chosen locations, presenting an aesthetics that engages in reconceptualizing centre and other and participates in the rhizomatic relations of a/r/tography" (p. 6, lines 213-17).

Author Response

-removed references to cultural capital

-added details re N'we Jinan in paper

-clarifications added

-questions addressed in paper

-The comment/question borders on the offensive. It is protocol in Canada at least to be as culturally specific as possible as a measure of Respect, which is one of the Seven Sacred/Grandfather teachings. Referencing culture follows the self-identification of the youth in the videos.

-a/r/t/ography reference removed

Reviewer 2 Report

Overall, I find this to be a compelling look at N'we Jinan's output, and the framing of it as a learning experience for the listener is a wise approach.  You clearly (and I believe accurately) identify space and spatial metaphors as central to N'we Jinan's work (line 221-222).

Where I have significant concern is in your methodological choice to use Inuit QaujimajatuqangitDene Kede, and Inuuqtigiit as frameworks when the examples introduced do not come from Inuit and Dene communities. In this way, the article risks a pan-Indigenous approach which runs counter to the specificity of learning about distinct communities offered by N'we Jinan.

Specific examples:

The introduction of Turtle Island in the abstract plays into this concern.  Turtle Island is a concept common across Canada, but does not belong to or exist in all Indigenous communities.  Inuit, for example, do not use the Turtle Island nomenclature, yet it is Inuit knowledge and values that form a significant part of the framework of this article.

In lines 73-74, each of these frameworks is introduced with the article 'the', which I feel is unnecessary.  'The' can essentialize and historicize Indigenous communities rather than reflect them (and their ideas) as current, varied, and dynamic.

Footnote 5, a continuation of the positionality statement, is broad to the point of being vague. My recommendation is to remove this footnote and add to the paragraph on positionality with language that describes the meaningful "connection with community" as an educator and in other ways.

I question the equivalencies of Dene Kede and Inuuqatigiit with Inuit Qaujimajatuqangit.  The former two documents are curriculum documents, while IQ is a governmental framework for development used in multiple departments - the documented cited is from the Nunavut Department of Education - but is not a curriculum document.  All play important roles, but each is not a "framework."

Line 262, introduces the Seven Sacred Teachings - but do not introduce their cultural source.  Again, these are not common to all Indigenous cultures in Canada, and certainly not Inuit, from whose culture two of your frameworks derive.

Footnote 11 explains that Cree lyrics are represented phonetically; was this the author's choice, or how N'we Jinan presents them?  This could be clearer.

Footnote 12 is better suited, in my opinion, to be in the body of the article.

On line 267, the author abbreviates IQ for the first time... why now and not earlier?

The author should be clear about which Inuktitut dialect is being used in the presentation of IQ values. Line 386, for example, introduces the IQ value of Interconnectedness.  The author introduces Aktuumanikkut Ilauniq as the Inuktitut translation for this, but this is not how the 2007 framework or the values posted in schools translate Interconnectedness, so I am curious as to the source.  I am familiar with Aktuauriunniqarniq (Kivalliq dialect) or Aktuaqattautiniq (Qiqiqtani dialect).

The accompanying footnote (22) describes the poster affiliated with interconnectedness, but is not clear that this is what is happening.  The phenomenon shown on the poster are sundogs.  Footnotes 13 and 14 are clear that they are describing the IQ poster images (though the note about clarifying dialect remains).  I only recognize this, however, because I am familiar with the images already.  It is not clear in the text where you draw this imagery from.

Footnote 22 could also be part of the body of the article rather than posed as a footnote. The inequal access to internet is more than a footnote when making the work of N'we Jinan accessible to Indigenous communities.

Overall, I feel this is a promising essay, but one that needs much more clarity in its methodological choices and which must provide better context for how IQ materials are being used.

Overall, the writing is clear and direct.  There are a few places that the voice changes from active to passive (such as line 36 or footnote #11) that hinder clarity.

The other comment I have regarding language is the use of youths vs youth.  Unless addressing a specific group of individual people, youth seems the more appropriate word choice here.  In the section from lines 156-162, "the youths" is repeated three times; enough to be distracting to a reader.

Also, Footnote #7 is numbered 77.

Author Response

-methodology clarified with accompanying examples which align with methodological clarifications

-all following points addressed/clarified in paper

Round 2

Reviewer 1 Report

This still reads like a student term paper – the first 1/3 jumps from jargony concept to jargony concept without fully describing or contextualizing any: digital storytelling, language activism, pluriversality, epistemic activism, etc.

I repeat my earlier comments as the author has not addressed these 2 major shortcomings:

1) Readers learn almost nothing about N’we Jinan other than they are “a group of young Indigenous artists who run a mobile production studio … [and] travel to different Indigenous communities” where they get Indigenous youth to perform in music videos. How many people comprise N’we Jinan? What genders? When was the organization formed? By whom, for what reasons? How is it organized? Is it a collective, an NGO, something else? We are led to believe that they are Cree – are they all Cree? Which Cree? What are their guiding principles? What are their funding arrangements? Are the youth paid for their performances? Do they receive training in video editing and production? Are the YouTube’s monetized? Who owns the copyrights to the videos? Where does N’we Jinan fit in the vast ecosystem of NGOs that purport to improve the lives of Indigenous youth?

2) How did the author collect data for the article? It appears that she watched the N’we Jinan YouTubes only. This is fine, but if that is all she did, she cannot make statements about what benefits the youth derive from participation. Did she interview members of N’we Jinan? Observe N’we Jinan’s work with any Indigenous youth? Speak to any youth who participated in a workshop?

It is not clear who the audience for the paper is. If Canadian educators are the audience, as the author seems to suggest in the first pages, Humanities is probably not the right vehicle. It is also not at all clear why the author describes her analysis as “autoethnographical” (p 3, line 64). First, the term is autoethnographic, not autoethnographical and it refers to a form of data collection/analysis/presentation in which the author draws on personal experiences and contexts to make points about broader social processes. Further, I see no evidence that this research for this paper was ethnographic (as claimed on p. 6, line 192) – the data seems to have been gathered by watching YouTube videos and consulting the N’we Jinan’s website. The author seems to be engaged in some form of discursive analysis, neither ethnography, nor autoethnography.

Footnote #16 – I can think of no reason why digital storytelling is necessarily “participatory” – It may be, but there is nothing inherent to a digital medium that creates the conditions for participation. The value of a digital medium is that the tools are widely accessible and easy to learn to use, making it accessible to diverse communities. How, for example, are audiences “invited to engage with Indigenous-created new media as learners rather than consumers”? (p. 7, line 257). How does the author know that viewers experience the videos as learners rather than consumers?

P. 4, lines 108-13: The author seems to be suggesting that the ills of colonialism: undrinkable water, inadequate housing, poor quality healthcare, etc. can be equated with/remedied by empowering Indigenous use to speak “their truths” by making music videos. This is absurd.

What is the basis for making a distinction between “Indigenous, Inuvialuit, and Metis youth” (p.4, lines 115-6)? Both Inuvialuit and Metis are Indigenous. Does the author mean to say First Nations when she writes “Indigenous”?

p.6, lines 175-176 – The phrase “Aklavik, never say die” originated in a specific context (unacknowledged by the author) in the 1950s and is credited to Arnold “Moose” Kerr, a non-Indigenous school principal. How have the Indigenous students described reinterpreted this local town motto?

Footnotes jump from #1 to #4 (2 and 3 are missing).

Author Response

Each of the comments have been addressed in the paper.

Reviewer 2 Report

I find that the author has engaged meaningfully with my earlier comments and improved the language to reflect the nuances involved in working with Indigenous communities.

I only note that there are gaps in footnote numbers (for example, it skips from 21 to 25 as I see the footnotes).

Author Response

Thank you for your helpful feedback. Comments have been addressed in the paper.

Round 3

Reviewer 1 Report

The author has not included a list specifying where and how previous comments were addressed.

Nonetheless, I see now that the only data for this paper are the author's viewing of the N'we Jiwan videos. While this is a legitimate form of literary analysis, these kind of data are insufficient to conclude that the videos "create a sense of connection in communities of people with Indigenous ancestry in Canada" or that the videos are a force for decolonization in Canada. The author can discuss the content of the videos or how educators might use them to decolonize their own practice, but the author has no information about any Indigenous or non-Indigenous people receive them and must not claim this.